# Epigenome-Wide Study Identified Methylation Sites Associated with the Risk of Obesity

**DOI:** 10.3390/nu13061984

**Published:** 2021-06-09

**Authors:** Majid Nikpay, Sepehr Ravati, Robert Dent, Ruth McPherson

**Affiliations:** 1Ruddy Canadian Cardiovascular Genetics Centre, University of Ottawa Heart Institute, 40 Ruskin St–H4208, Ottawa, ON K1Y 4W7, Canada; 2Plastenor Technologies Company, Montreal, QC H2P 2G4, Canada; ravatisepehr@gmail.com; 3Department of Medicine, Division of Endocrinology, University of Ottawa, the Ottawa Hospital, Ottawa, ON K1Y 4E9, Canada; bdent@toh.ca; 4Atherogenomics Laboratory, University of Ottawa Heart Institute, Ottawa, ON K1Y 4W7, Canada

**Keywords:** obesity, EWAS, epigenetics, multiomics, Mendelian randomization

## Abstract

Here, we performed a genome-wide search for methylation sites that contribute to the risk of obesity. We integrated methylation quantitative trait locus (mQTL) data with BMI GWAS information through a SNP-based multiomics approach to identify genomic regions where mQTLs for a methylation site co-localize with obesity risk SNPs. We then tested whether the identified site contributed to BMI through Mendelian randomization. We identified multiple methylation sites causally contributing to the risk of obesity. We validated these findings through a replication stage. By integrating expression quantitative trait locus (eQTL) data, we noted that lower methylation at cg21178254 site upstream of *CCNL1* contributes to obesity by increasing the expression of this gene. Higher methylation at cg02814054 increases the risk of obesity by lowering the expression of *MAST3*, whereas lower methylation at cg06028605 contributes to obesity by decreasing the expression of *SLC5A11*. Finally, we noted that rare variants within 2p23.3 impact obesity by making the cg01884057 site more susceptible to methylation, which consequently lowers the expression of *POMC*, *ADCY3* and *DNAJC27*. In this study, we identify methylation sites associated with the risk of obesity and reveal the mechanism whereby a number of these sites exert their effects. This study provides a framework to perform an omics-wide association study for a phenotype and to understand the mechanism whereby a rare variant causes a disease.

## 1. Introduction

Obesity is a complex phenotype and the outcome of numerous genes and environmental factors. Epigenetic sites are considered as the sites of gene–environment interactions. Epigenetics provides an elegant solution to modify gene expression activity in response to external stimuli without altering the DNA code. As such, it has an important role in the regulation and manifestation of complex phenotypes.

The relation between epigenetics and obesity has been the subject of numerous studies over the past few years [1,2]. Methodological improvements, and the global increase in obesity, have contributed to this interest. The general consensus from these studies is that interindividual variation in epigenetic modifications correlates with body weight. Furthermore, findings from these studies support not only a role for epigenetics in gaining weight, but also epigenetic alterations as a response to obesity [1,2,3,4,5,6].

As reviewed by Ling et al. [1], the majority of previous studies had small sample sizes, which lowers the power of statistical tests. Furthermore, they typically measured both epigenetic levels and BMI in the same group of individuals and then investigated the epigenetic sites that showed differential levels of modifications in individuals with a higher BMI than those with a lower BMI; however, such a design cannot tell us whether a significant association indicates causation (e.g., Methylation at a site → Obesity), correlation (Methylation ← confounders → Obesity) or reverse causation (Obesity → Methylation).

To overcome these issues, in this study, we used Mendelian randomization (MR) that can control for both confounding and reverse causation. MR is a form of instrumental variable analysis that investigates the relationship between the exposure (DNA methylation) and the outcome (BMI) using an instrument (a set of independent SNPs) that is known to cause change in the exposure. Alleles of independent SNPs are randomly allocated to offspring at conception (Mendel’s second law); therefore, an instrument of SNPs is inherently immune to the confounding effect of environmental factors that can bias an association study. Furthermore, by excluding SNPs with pleiotropic effect (Methylation ← SNP → Obesity) from the instrument, it is possible to rule out the correlation scenario, and by swapping the places of exposure and outcome and repeating the test, we can investigate the possibility of reverse causation.

To increase the statistical power of our analyses, in this study, we used a two-sample MR design (Figure 1) that incorporated data from separate studies to estimate a causal effect of the exposure on the outcome. Hence, with this design, we can achieve a better statistical power by including data from the GWAS consortia [7].

Examining the association between change in DNA methylation at every site in the genome and obesity using MR is cumbersome. Therefore, in this study, we used a SNP-based multiomics pipeline that also included MR, in order to efficiently narrow down our search and identify methylation sites that contribute to the risk of obesity. We validated our findings through the replication stage; we also investigated the mechanism whereby these sites contribute to obesity by integrating eQTLs data into our analysis.

## 2. Methods

Our SNP-based multiomics design was previously described [8,9]. In summary, our approach requires full GWAS summary statistics as input files and a genotype file to calculate the linkage disequilibrium (LD) between SNPs. In this study, we used mQTL data from McRae et al. [10] and GWAS data for BMI from the meta-analysis of the UK Biobank (UKBB) and Giant consortium [11] to identify methylation sites that contribute to the risk of obesity. We used the genotype data from the 1000 genomes (*n* = 503 of European ancestry) to calculate the LD between SNPs.

Initially, we tested whether SNPs that are associated with BMI co-localize with mQTLs for a methylation site. Co-localization analysis was done using the SMR software (version 1.03) [12] that can also differentiate between a pleiotropic effect (its null hypothesis) and a linkage effect through the HEIDI test. Methylation sites that their QTLs co-localize with BMI risk SNPs (with P_SMR_ < 5 × 10^−8^ and P_HEIDI_ > 0.05), were then subjected to Mendelian randomization analysis. Note that P_SMR_ indicates the *p*-value from the co-localization test and P_HEIDI_ represents the *p*-value from the pleiotropy test. Mendelian randomization (MR) aims to investigate the relationship between the exposure (methylation at a site) and the outcome (BMI) using a set of independent SNPs (instrument). SNPs included in the instrument must meet a number of criteria.

(1)They must not be in LD. In this study, we used SNPs that were in linkage equilibrium (r^2^ < 0.05).(2)They must not show a pleiotropic effect (i.e., Exposure ← SNP→ Outcome). We excluded such SNPs from the instrument by using (P_HEIDI_ < 0.01).(3)They must be significantly associated with exposure; we used SNPs that are associated with exposure at the GWAS significance level (*p* < 5 × 10^−8^).

We did the MR analysis using the GSMR algorithm implemented in GCTA software (version 1.92) [13]. As compared to other methods for MR analysis, GSMR automatically detects and removes SNPs that have a pleiotropic effect on both exposure and outcome; in addition, it accounts for the sampling variance in beta estimates and the LD among SNPs; as such, it is statistically more powerful than other MR approaches [13].

The GSMR algorithm examines the contribution of exposure on outcome by extracting summary association statistics (Beta, SE) for SNPs included in the instrument from the input files. It then generates a scatter plot by contrasting effect sizes (betas) of SNPs on exposure with their corresponding effect sizes on outcome and calculates the slope (β) of the line of best fit and variance around it using generalized least squares regression. In this context, a significant +β indicates subjects that are genetically susceptible to higher levels of exposure (e.g., methylation at a site), tend to have higher risk of outcome (e.g., obesity).

To investigate the possibility of reverse causation or the scenario that a change in the methylation level at a site is merely the consequence of gaining weight (BMI → Methylation site), we selected methylation–BMI pairs with *p* < 5 × 10^−8^ from our MR analysis and swapped the places of exposure and outcome. Namely, we set BMI as the exposure and the methylation probe as the outcome, and then, we re-performed the MR test and excluded any methylation–BMI pairs with significant evidence of reverse causation (*p* < 0.05).

## 3. Results

Through our analysis pipeline, we integrated mQTL association summary statistics with GWAS summary statistics for BMI to identify the methylation sites that contribute to the risk of obesity. We identified multiple sites that are causally associated with BMI (Appendix A). We replicated (Appendix A) these associations using data from Hannon et al. [14,15]. Next, we integrated the transcriptome data from the eQTLGen consortium [16] to investigate the molecular mechanism whereby these sites impact obesity. We review the main findings below.

### 3.1. CCNL1 Locus

Co-localization analysis revealed mQTLs for cg21178254 methylation site upstream of *CCNL1* gene co-localize (Appendix A) with obesity risk SNPs (P_SMR_ = 7 × 10^−10^, P_HEIDI_ = 0.5). The top SNP rs62274156-T in this region was associated with increased BMI (B = 0.01, *p* = 3 × 10^−10^) and lower methylation at the cg21178254 site (B = −0.96, *p* = 5 × 10^−182^). Consistently, the MR analysis revealed that lower methylation at this site contributes to a greater risk of obesity (B = −0.02, *p* = 1.2 × 10^−9^, Appendix A).

Next, we investigated whether methylation at the cg21178254 site impacts the expression of nearby genes by performing a MR analysis. Regional association plots for mQTLs of cg21178254 and eQTLs of the *CCNL1* gene were found to overlap (Figure 2A). The MR analysis revealed that as this site becomes methylated, the expression of the *CCNL1* gene decreases (β = −0.4, *p* = 5 × 10^−143^; Figure 2B) and this contributes to BMI (β = 0.04, *p* = 2 × 10^−10^; Figure 2B). The expression of no other gene within the 3q25 chromosome band was affected by methylation of the cg21178254 site. As such, it appears methylation at this site contributes to obesity by changing the expression of *CCNL1*.

### 3.2. SLC5A11 Locus

We found mQTLs for the cg06028605 site within the *SLC5A11* gene overlap with obesity risk SNPs (P_SMR_ = 2.6 × 10^−10^, P_HEIDI_ = 0.3). The top SNP rs34172679-T in this region was associated with a lower BMI (B = −0.02, *p* = 2 × 10^−10^) but higher methylation at the cg06028605 site (B = 1.1, *p* = 1 × 10^−209^). Consistently, MR analysis revealed that lower methylation at this site contributes to a higher BMI (B = −0.01, *p* = 3.3 × 10^−9^, Appendix A).

The regional association plots for mQTLs of the cg06028605 site, eQTLs of *SLC5A11* and BMI risk SNPs overlap (Figure 3A). MR analysis revealed that this site contributes to the risk of obesity by changing the expression of the *SLC5A11* gene. We found that, as this site becomes methylated, the expression of *SLC5A11* increases (B = 0.5, *p* = 3.6 × 10^−194^, Figure 3B) and this contributes to a lower BMI (B = −0.03, *p* = 2.8 × 10^−11^, Figure 3B).

### 3.3. MAST3 Locus

Co-localization and subsequent MR analysis revealed that the cg02814054 methylation site within *MAST3* contributes to obesity. We noted that the regional association plots for cg02814054 mQTLs, *MAST3* eQTLs and BMI risk SNPs overlap (Figure 4A). The top mQTL, rs4808745-T, within this region was associated with higher methylation (B = 0.96, *p* = 1.4 × 10^−200^), lower expression of *MAST3* (B = −0.32, *p* = 2.7 × 10^−269^) and a higher BMI (B = 0.02, *p* = 7 × 10^−14^, Appendix A). Concordantly, MR analysis revealed that, as this site becomes methylated, the expression of *MAST3* decreases (B = −0.2, *p* = 1.6 × 10^−69^, Figure 4B) and the risk of obesity increases (B = −0.06, *p* = 6.9 × 10^−14^, Figure 4B).

### 3.4. Rare Variants in 2p23.3

Finally, we examined whether rare obesity SNPs exert their effect through an epigenetic site. For this purpose, we obtained the list of rare variants for obesity from the ClinVar database [17] (Appendix A) and matched their position with the position of mQTLs. We noted that rare SNPs within the 2p23.3 chromosome band overlap with mQTLs (blue circles) for a methylation site (cg01884057) within this locus (Figure 5A). MR analysis revealed the higher methylation at this site was associated with a higher BMI (B = 0.6, *p* = 1.9 × 10^−60^, Figure 5B). We confirmed this finding using data from a second study as well (B = 0.03, *p* = 1.8 × 10^−61^, Table 1). Higher methylation at this site was associated with lower expression of *POMC* (B = −0.11, *p* = 1.75 × 10^−37^), *ADCY3* (B = −0.08, *p* = 4.3 × 10^−20^) and *DNAJC27* (B = −0.1, *p* = 5.8 × 10^−30^, Table 1). Furthermore, the lower expression of these genes was associated with a higher risk of obesity (Table 1), consistent with the established role of these genes in obesity. Thus, rare variants within this locus impact obesity by making this site susceptible to methylation, which consequently attenuates the expression of *POMC*, *ADCY3* and *DNAJC27*.

## 4. Discussion

Here, we performed a genome-wide search for methylation sites that contribute to obesity. Through a discovery and replication stage, we identified seven sites that are causally associated with the risk of obesity. Epigenetic sites are genomic regions that undergo chemical modifications in response to environmental factors; however, the underlying sequence of the DNA also has a role. Namely, some combination of alleles makes a site more susceptible to epigenetic modification than the others. Therefore, our results should be considered in this context.

We integrated the transcriptome data from the eQTLGen consortium [16] to investigate the likely mechanism whereby the methylation sites impact obesity. We found that lower methylation at the cg21178254 site upstream of *CCNL1* contributes to obesity by increasing the expression of this gene. We also examined the data from the PhenomeXcan database [18], which is a comprehensive repertoire of gene expression–trait associations. Results from this database also indicated that a higher expression of this gene is associated with a higher risk of BMI (*p* = 1.4 × 10^−6^). Previous studies found that SNPs upstream of *CCNL1* are associated with leptin levels [19] and birth weight [20]. We also noted that a change in expression of this gene is associated (*p* < 1 × 10^−10^) with comparative height at age 10 as well as standing height; therefore, this gene may impact anthropometric traits including BMI by a change in transcriptional activity early in life. This is consistent with the function of this gene, which is a cyclin-dependent kinase and is involved in regulating transcription and mRNA processing.

We noted that the lower methylation at the cg06028605 site in the *SLC5A1*1 locus contributes to obesity by decreasing the expression of this gene. Data from PhenomeXcan indicated that the lower expression of this gene is associated with a higher BMI (*p* = 1 × 10^−7^). SLC5A11 is involved in the transport of glucose and other sugars, bile salts, organic acids, metal ions and amine compounds. The gene is known to have a regulatory impact on appetite. Food deprivation is reported to increase the expression of *SLC5A11* and the excitability of SLC5A11-expressing EB R4 neurons [21]. Ugrankar et al. [22] reported that SLC5A11 is involved in glucose-regulated feeding and may function downstream of insulin signaling on the pathway of glucose sensing.

We found that the higher methylation at the cg02814054 site within *MAST*3 contributes to obesity by lowering the expression of this gene. Data from PhenomeXcan also revealed that the lower expression of this gene contributes to a higher BMI (*p* = 1 × 10^−8^). *MAST3* is a microtubule-associated serine/threonine-protein kinase. The gene appears to have diverse activities and is implicated in different diseases. Therefore, the molecular path whereby a change in expression of this gene contributes to obesity remains elusive. Data from PhenomeXcan (based on GTEX and UKBB data) indicated that *MAST3* has a higher expression in people with a college or a university degree (*p* = 7.2 × 10^−7^). We also confirmed this effect through MR analysis (B = 0.03, *p* = 5.8 × 10^−9^, using data from eQTLGen and UKBB). Therefore, the mechanism whereby a change in expression of this gene contributes to obesity may be through the neural paths. This is reasonable considering that CNS contributes to obesity and a number of studies have documented the neural function of this gene [23,24].

Over the past decades, GWAS studies have identified SNPs that contribute to phenotopic variability, and high-throughput studies have identified SNPs underlying functional elements. With the availability of data from these studies, and analytical tools that rely on summary association statistics, now is the time to integrate findings from these studies and catalog the association of functional elements with the phenome. In this work and previously [8,9,25], we showed such studies are feasible through a SNP-based multiomics approach. Furthermore, in this study, we show that such an approach can be used to understand the function of rare genomic variants (Figure 5). Although advances in sequencing technology allow us to capture rare variants efficiently, it often remains elusive which variant has a causal impact on the examined phenotype. We demonstrate that, by leveraging the abundance of knowledge available at nearby common SNPs, it is possible to investigate the mechanism whereby a rare SNP impacts a phenotype. Such an approach also provides the possibility to identify the causal rare SNP in a list of candidate SNPs.

This study is notable considering that it combines several sets of publicly available data to reveal new findings; therefore, it demonstrates the value of data sharing by researchers and supports this practice. Compared to traditional EWAS studies, this study is unique because it exploits the power of MR to control for confounding and reverse causation; as such, it provides a framework for performing omics-wide association studies of phenotypes. Our study has a number of limitations; in this study, we could not reveal any functional insight for a number of the identified sites (Appendix A). Therefore, future studies require integrating a more diverse and comprehensive set of data in order to investigate the mechanism whereby a functional element impacts a phenotype. Another limitation of this study is that we could not investigate trans-regulatory effects since the majority of QTL studies (including those examined in this study) report their findings for cis-acting QTLs (e.g., eQTLs within a gene’s own regulatory region). Reporting QTLs with trans-regulatory effects (e.g., eQTLs located in a different chromosome than their gene) is highly valuable because they can provide novel insights into the molecular pathway whereby a biomarker exerts its effect.

In summary, in this study, we integrated several sets of data to identify epigenetic sites that contribute to obesity. For a number of these sites, we revealed the mechanism through which they impact obesity. Finally, we showed that, by leveraging the knowledge of common SNPs, it is possible to prioritize rare variants and investigate the mechanism whereby a causal variant impacts a phenotype.

## Figures and Tables

**Figure 1 nutrients-13-01984-f001:**
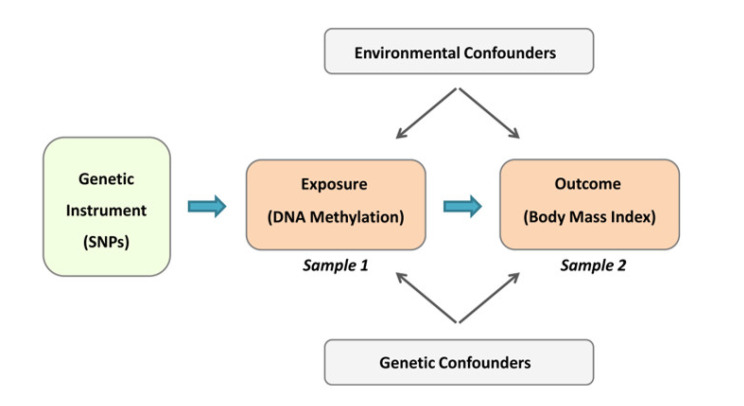
**Concept of two-sample Mendelian randomization (MR) design.** Under this design, we first constructed an instrument based on a set of independent SNPs that were significantly associated with the exposure in the first sample. Next, we contrasted effect sizes (betas) of SNPs on the exposure with their corresponding effect sizes on the outcome (obtained from the second sample) to find out if there was a significant association. This design is immune to confounding environmental factors because it uses SNPs as instrument. Furthermore, by removing SNPs with peliotropic effect, genetic confounding is prevented. Further information is provided in the Methods section.

**Figure 2 nutrients-13-01984-f002:**
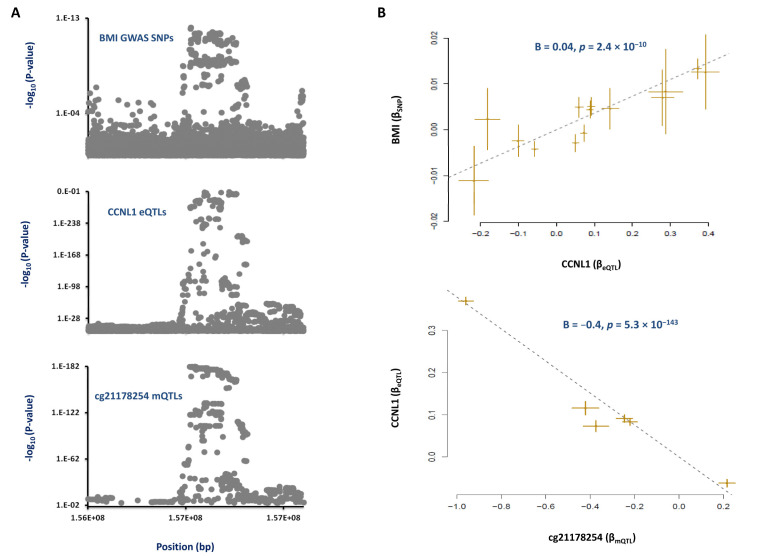
**Mechanism whereby the cg20892840 site upstream of *CCNL1* impacts obesity.** (**A**) We found that regional association plots for mQTLs of cg20892840, eQTLs of *CCNL1* and risk SNPs of obesity co-localize. (**B**) MR analysis then revealed that, as cg20892840 becomes hypomethylated, the expression of *CCNL1* increases, and this contributes to a higher BMI. Each point represents a SNP; the x-value of a SNP is its β effect size on the exposure, and the horizontal error bar represents the standard error around the β. The y-value of the SNP is its β effect size on the outcome, and the vertical error bar represents the standard error around its β. The dashed line represents the line of best fit (a line with the intercept of 0 and the slope of β from the MR test).

**Figure 3 nutrients-13-01984-f003:**
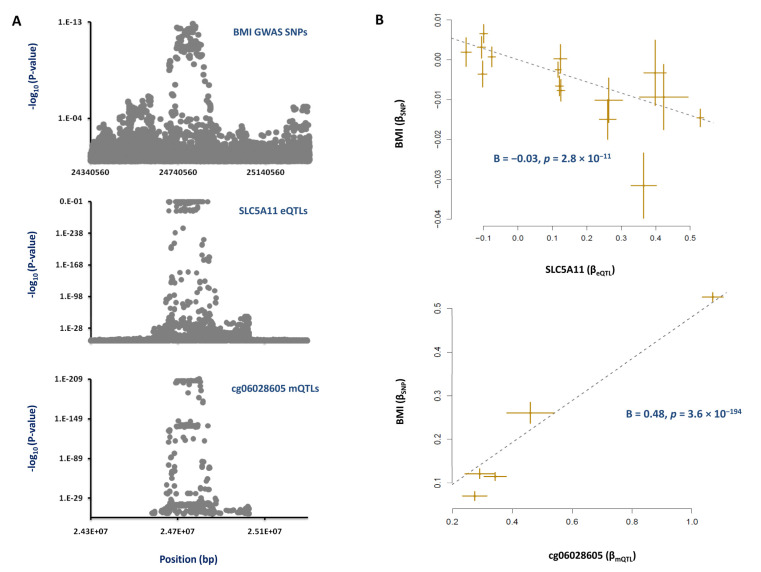
**Mechanism whereby the cg06028605 site inside *SLC5A11* impacts obesity.** (**A**) We found that the regional association plots for mQTLs of cg06028605, eQTLs of *SLC5A11* and risk SNPs of obesity co-localize. (**B**) MR analysis then revealed that, as the cg06028605 site becomes hypermethylated, the expression of *SLC5A11* increases, and this contributes to a lower BMI. Each point on the MR plots represents a SNP; the x-value of a SNP is its β effect size on the exposure, and the horizontal error bar represents the standard error around the β. The y-value of the SNP is its β effect size on the outcome, and the vertical error bar represents the standard error around its β. The dashed line represents the line of best fit (a line with the intercept of 0 and the slope of β from the MR test).

**Figure 4 nutrients-13-01984-f004:**
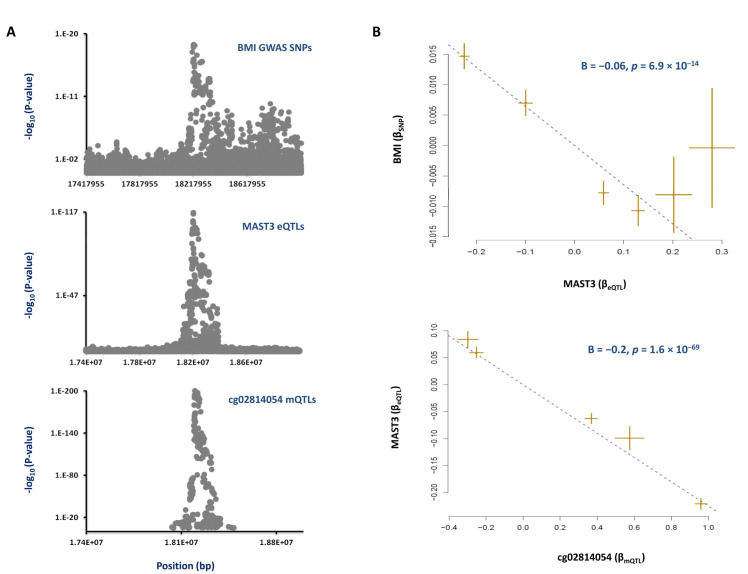
**Mechanism whereby the cg02814054 site inside *MAST3* impacts obesity.** (**A**) We found that the regional association plots for mQTLs of cg02814054, eQTLs of *MAST3* and risk SNPs of obesity co-localize. (**B**) MR analysis then revealed that, as the cg02814054 site becomes hypermethylated, the expression of *MAST3* decreases, and this contributes to a higher BMI. Each point on MR plots represents a SNP; the x-value of a SNP is its β effect size on the exposure, and the horizontal error bar represents the standard error around the β. The y-value of the SNP is its β effect size on the outcome, and the vertical error bar represents the standard error around its β. The dashed line represents the line of best fit (a line with the intercept of 0 and the slope of β from the MR test).

**Figure 5 nutrients-13-01984-f005:**
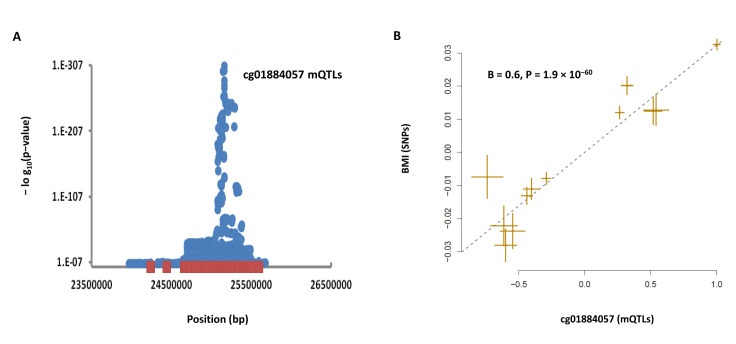
**Understanding the function of rare variants using the available knowledge on nearby common SNPs.** In this example, we noted that (**A**) rare variants (red squares) within the 2p23.3 locus overlap with mQTLs (blue circles) for a methylation site (cg01884057) within this locus. (**B**) MR analysis revealed that higher methylation at this site was associated with a higher BMI. Methylation at this site was also associated with lower expression of *POMC*, *ADCY3* and *DNAJC27* (Table 1). Therefore, rare variants within this locus impact obesity by making this site more inclined to methylation, which consequently lowers the expression of *POMC*, *ADCY3* and *DNAJC27*. Each point on MR plots represents a SNP; the x-value of a SNP is its β effect size on the exposure, and the horizontal error bar represents the standard error around the β. The y-value of the SNP is its β effect size on the outcome, and the vertical error bar represents the standard error around its β. The dashed line represents the line of best fit (a line with the intercept of 0 and the slope of β from the MR test).

**Table 1 nutrients-13-01984-t001:** Summary association statistics for the mechanism whereby the cg01884057 site contributes to obesity.

Biomarker	PMID	Oucome	PMID	Beta	SE	P	NSNP
cg01884057	30514905	*POMC* expression	bioRxiv 447367	−0.11	0.01	1.7 × 10^−37^	5
cg01884057	30401456	*POMC* expression	bioRxiv 447367	−1.90	0.15	3.6 × 10^−37^	4
*POMC* expression	bioRxiv 447367	BMI	30239722	−0.03	0.01	1.5 × 10^−10^	13
cg01884057	30514905	*ADCY3* expression	bioRxiv 447367	−0.08	0.01	4.3 × 10^−20^	12
cg01884057	30401456	*ADCY3* expression	bioRxiv 447367	−1.59	0.16	5.5 × 10^−24^	5
*ADCY3* expression	bioRxiv 447367	BMI	30239722	−0.06	0.01	2.2 × 10^−10^	5
cg01884057	30514905	*DNAJC27* expression	bioRxiv 447367	−0.10	0.01	5.8 × 10^−30^	6
cg01884057	30401456	*DNAJC27* expression	bioRxiv 447367	−1.56	0.14	2.3 × 10^−27^	4
*DNAJC27* expression	bioRxiv 447367	BMI	30239722	−0.05	0.01	2.7 × 10^−10^	7

## Data Availability

Summary association statistics to replicate our findings are provided in the Appendix A. Samples scripts and input files to perform a SNP-based multiomics analysis are available from: https://github.com/mnikpay/Multiomics-MR-scripts.git (accessed on 8 June 2021).

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
