# Peer review of "Epigenome-Wide Study Identified Methylation Sites Associated with the Risk of Obesity"

_nutrients, 2021, doi:10.3390/nu13061984_

Round 1
Reviewer 1 Report
The manuscript entitled “Epigenome wide study identified methylation sites associated with the risk of obesity” by Nikpay et al. performed an epigenome-wide study to identify methylation sites associated with higher BMI. The authors integrated the results of EWAS with eQTL reports and found that three candidate regions of the risk of obesity. The authors reported that the multi-omics approach would be a powerful tool to identify the genomic regions for the risk of obesity. It is an interesting study that reports the genomic regions with the risk of obesity; however, several essential points need to be addressed.
Major concerns:
1. The authors claimed that “the methylation sites are causally contributing to the risk of obesity”; however, the authors did not test if these methylation alterations cause the obesity, the authors performed association studies. Therefore, the results of the study do not support this claim.
2. The authors reported these mQTLs and eQTLs of the genes were overlapped; therefore, the methylation status due to the genetic changes are associated with expression alterations. However, these associations were observational, and the connections were not clear in this study. The authors need to discuss how these alterations affect the expression of the genes. Moreover, 1) location of the mQTL/eQTL to the TSS of the gene, 2) the allele differences of methylation and expression status (both homozygous and heterozygous) at least the mQTL with the highest association should be included in the figures.
3. Page 7, line 198-201, the authors discussed, “In this study, we also noted higher expression of this gene in people with a college or a university degree (B=0.03, 199 P=5.8e-9), highlighting the neural function of this gene. Therefore, the mechanism whereby change in expression of this gene contributes to obesity may be through the neural paths.”
The authors need to clarify how this analysis was performed. For example, if the authors tested that obesity status is associated with education levels, the authors tested the association using multivariate analyses (the covariates in the models should be indicated).
Minor concerns:
All Figures should have axis labels.
Reviewer 2 Report
Although the topic is interesting, the Authors should improve all the sections of this manuscript. The introduction needs extension and a description of previous research. Methods should be described better. The results section is probably the one which probably needs less revision. The discussion should present strenght and weakness of this study.
Round 2
Reviewer 2 Report
The authors have imptoved their manuscript according to my previous suggestions. However, I have noted that, in the Introduction section, the authors did not include any references from previous studies. In my opinion, the introduction section should describe the backgroud and rationale of a study. For this reason, the authors are encouraged to provide references of previous studies.
